# Qualitative Research Studies Addressing Patient-Practitioner Communication about Online Health Information

**DOI:** 10.3390/ijerph192114004

**Published:** 2022-10-27

**Authors:** Cathrin Brøndbo Larsen, Heidi Gilstad

**Affiliations:** 1Department of Language and Literature, Norwegian University of Science and Technology, 7491 Trondheim, Norway; 2Department of Neuromedicine and Movement Sciences, Norwegian University of Science and Technology, 7491 Trondheim, Norway; 3Department of Rheumatology, St. Olavs University Hospital, 7030 Trondheim, Norway

**Keywords:** eHealth literacy, online health communication, internet informed patients, patient-practitioner communication

## Abstract

Modern healthcare is becoming increasingly technologized, knowledge-intensive, and specialized, which has consequences for communication between patients and practitioners in clinical encounters. Health information is increasingly accessible to patients through online resources. The objective of this paper is to provide an overview of existing studies that address how patients communicate with practitioners about online health information and to identify the key topics raised in these studies. With the concept of eHealth literacy as its point of departure, this paper specifically highlights the eHealth literacy topic of how patients *comprehend*, *appraise* and *communicate* online health information before and during the encounter with the healthcare professionals. In the literature search, we focus on qualitative studies that consider patients’ and/or practitioners’ reflections on online health information. We searched PUBMED, SCOPUS and Web of Science to capture research from various fields. Sixteen studies were included that met the following criteria: Qualitative study, focus on patient-practitioner communication, eHealth literacy and online information. The results show that the vast majority of studies were based on qualitative interviews, addressing patients’ and practitioners’ perspectives. Key topics in studies addressing patient perspectives were: *reasons for seeking online information*; *calibrating understanding* of online information with the practitioner; and *barriers to discussing online information* with practitioners. Key topics raised in studies focusing on practitioners were: *trust* in the patient and the online health information he/she presented as well as *strategies to respond to patients* who referred to online health information. The review highlights the need for further qualitative studies, preferably observational studies from authentic clinical communication situations, in order to capture how patients and practitioners communicate about online health information.

## 1. Introduction

Modern health services are increasingly digitalized and specialized. Various digital solutions are available that enable patients to take responsibility and control of their own health [1]. Patients need to adapt to more advanced technologies and growing amounts of available health information and knowledge. Large amounts of potentially useful knowledge, technologies and methods result in many decision points in a patient’s pathway, which in turn creates ever increasing demands on information and communication to ensure fundamental patient rights such as the right to know, decision-making and ethical issues of privacy.

The continuous development of the health services and the health discourses (politics, rules, regulations, professional practices) has an impact on the situated communication in clinical encounters. Patients often present in the clinic well informed about their condition, and in addition to having a physical examination, they want to discuss their understanding of what they have read online with the healthcare professional’s knowledge [2]. Sufficient usage of eHealth tools is also a challenge for some patients [3]. Consultations with healthcare practitioners can be an opportunity for patients to learn and develop their competencies [2,4,5].

Studies concerned with how patients’ access, interpret and apply health information to solve health issues are often referred to as (e)Health literacy studies. The field has evolved from pioneers *defining the concepts* Health literacy [6,7] and eHealth literacy (eHL) [8,9] to *focusing on how to measure* (e)Health literacy on a population level.

In this paper, we want to draw attention to the interplay between eHealth literacy and the social context. We are more specifically interested in identifying existing studies about how patients informed by online health information communicate about this with health care practitioners. First, we offer an introduction to the key concepts: health literacy, eHealth literacy and context. 

### 1.1. Health Literacy and eHealth Literacy

The concepts “*health literacy*” and “*eHealth literacy*” are often used interchangeably, with the distinction that the latter involves digital technologies [2]. Health literacy (HL) has long been established as a concept in healthcare literature. [6,7] Don Nutbeam connected HL to public health promotion and established it as a critical means of patient empowerment. He later outlined three levels of health literacy: Functional HL (traditional literacy and information on health risks, and use of health care systems), Interactive HL (development of personal skills and improved personal capacity to act independently on knowledge) and Critical HL (cognitive skills oriented towards social and political actions and individual actions) [7]. The levels highlight the individual’s cognitive skills of perceiving, interpreting and communicating health information critically in the context. Sørensen and colleagues [10] developed the conceptual framework and suggested a new definition of HL “*the knowledge, motivation and competences to access, understand, appraise and apply health information in order to make judgments and take decisions in everyday life concerning health care, disease prevention and health promotion to maintain or improve quality of life throughout the course of life*.” [10]. This definition suggests a perspective on the subject as actively making health decisions throughout one’s life.

Although the concept of health literacy has evolved alongside societal development, some researchers thought it prudent to adapt the terminology to the digital world in response to the increasing digitization of public health communication. The term eHealth literacy (eHL) was introduced by Norman and Skinner [8] to describe the complexity of competencies needed for health communication in the digitalized modern society. With the Lily model, they illustrated six main components of eHL: basic literacy (reading, writing, numeracy), information literacy, media literacy, computer literacy, scientific literacy, and HL. Griebel et al. conducted a review of the current literature on eHL [2], added socio-contextual and technological aspects to the understanding of eHL, and defined eHL as follows: *“eHealth literacy includes a dynamic and context-specific set of individual and social factors, as well as consideration of technological constraints in the use of digital technologies to search, acquire, comprehend, appraise, communicate, apply and create health information in all contexts of healthcare with the goal of maintaining or improving the quality of life throughout the lifespan*” (p. 17). Unlike the previous definitions of HL and eHL, this definition explicitly linked eHL aspects to the context by underlining the dynamic and context-specific preconditions for access, interpretation and use of health information.

In this paper we are specifically interested in the eHL topic: how patients comprehend, appraise and communicate online health information, before and during the encounter with the healthcare professional. The paper adopts the understanding of eHL as being closely related to the social contexts in which it is realized.

### 1.2. Context

In a dialogic approach on language and communication, meaning making and interpretation relate to different *levels of context*. Linell (1998:128) [11] distinguishes between two levels of contextual resources: the *immediate* and the *mediate*. The immediate contextual resource is the concrete physical setting; for example, the office in which the clinical encounter is taking place. The mediate contexts are more abstract contextual resources, for example: what participants assume, believe, know or understand about the things talked about in the discourse; the current and upcoming communicative projects of the participants; specific knowledge or assumptions about persons involved, the abstract situation definition; the organizational and the socio-historically constituted contexts of institutions and subcultures; the knowledge of language, communicative routines and action types, and the general background knowledge.

In this perspective, context is a dynamic phenomenon, where the participants in the communication situation actively co-create the context in the meaning-making process. This co-creation of meaning is influenced by the roles and responsibilities of the participants, the organizational and professional rules and norms, and the knowledge and experiences of the participants.

### 1.3. Objective and Aim of the Study

The objective of this paper is to get an overview of qualitative studies addressing how patients informed by online health information communicate about this with health care practitioners, and to identify the key topics raised in these studies. The research questions guiding this paper are:In the selected studies, which topics are raised concerning online health information in the communication between health care professionals and patients?How do these topics connect with the eHealth literacy aspect of how patients comprehend, appraise and communicate online health information?

## 2. Materials and Methods

A systematic approach was adopted to provide an overview of the literature. The authors decided on strategy and search words before proceeding with the search in June 2021. Various search words and sites were tested. Eventually three search monitors—PubMed, Web of Science, SCOPUS—were selected and deemed sufficient to cover the various fields. The following search words were extended to all the search monitors:

Qualitative research OR qualitative study OR qualitative methods) AND (Literacy) AND (health communication OR patient communication OR patient participation OR consultation OR patient education OR physician-patient relations OR nurse-patient relations OR practitioner-patient relations) AND (eHealth OR e-health OR digital health OR Internet information OR online information).

### 2.1. Data Handling

Before the data set was ready, 260 duplicates were removed in Endnote. We were left with 1036 articles, which were screened using the RAYYAN software tool. Through screening of the headlines and abstracts, we picked 52 articles which were read more closely before 16 studies (see Figure 1) were eventually included which met the following criteria: Qualitative study, focus on patient-practitioner communication, and eHealth literacy/internet information. As we wanted to get an overview of the field in general, we wanted to include research from all countries, patient groups and health care settings. We included articles that were older than 2002, aligning with the beginning of the WWW 2.0 era. We focused on the adult population (18 and above).

#### 2.1.1. Inclusion Criteria


Qualitative researchFocus on patient-practitioner communication,Focus on eHealth literacy/internet information.Research published in English, Norwegian, Danish, or Swedish


#### 2.1.2. Exclusion Criteria


Research method (surveys, review, randomized control trials, cohorts, interventions)Non-research documentsResearch that was based on simulated conversationResearch that was aimed towards development or testing of various toolsFocus on intermediated communication with a translator


## 3. Findings

The review included qualitative studies addressing online health information in the communication between practitioners and patients (see Appendix A for overview). We wanted to include both observational studies and qualitative interviews of practitioners and/or patient. However, our search resulted in mostly interview-based studies, including focus group discussions. One article had data from both interviews and observations [13]. However, as the authors reported that they found no talk about online health information in the observational data, the results are focused upon the interview data. Ten studies addressed *patient perspectives* on the topic of online information in the communication between health personnel and patients [13,14,15,16,17,18,19,20,21,22,23,24] in different medical areas: Patients with chronic disease [17,18,19], or gynecologic or breast cancer [15]; senior citizens [16,23]; parents or mothers to be [21,24]. Additionally, four articles addressed the *practitioners’ perspective* [21,23,25,26,27] and two articles described interviews from both patients and practitioners’ perspective [13,17]. In the results section, we will present a descriptive overview of the main findings from the patients’ perspectives and the health care practitioners’ perspectives, respectively, on the topic of online health information.

### 3.1. Patients’ Perspectives on Online Health Information

The 12 studies with a patient perspective were semi-structured interviews of patients in various populations. The articles also reported many findings regarding general online information attitudes and patients’ information-seeking habits. In this article we have focused upon results that address online health information in patient-practitioner communication. A systematic reading of the articles, where the topics were coded and categorized thematically, revealed the following recurring topics: (1) Reasons to seek online information, (2) calibrating one’s own understanding of online health information with the practitioner, (3) barriers to discussing online information with practitioners. In the following section, we elaborate in the three recurring topics in the articles with a patient perspective.

#### 3.1.1. Reasons to Seek Online Information

In the studies, the informants reported different reasons for searching health information online. In the multiple-country study by Diviani et al. [14] reasons mentioned were self-diagnosis or to gather enough information to avoid the doctor’s office altogether. Others used to search the internet prior to, or after the medical consultation, to meet prepared, or to complement the practitioner’s information, and on rare occasions, to challenge the doctor. Jiang et al., (2022) also reported distrust in practitioners as a reason why patients searched the internet for more information, especially of there was a lack of information, or a hope of wonder medicines [17]. However, in Lee’s study, patients’ main reasons for online searches were to be more informed and gain knowledge and understanding about their disease. Some patients sought information to find answers the doctor could not give them, or to avoid long waiting lists [18]. Several studies reported that searching the internet for health-related causes was a way of avoiding unnecessary visits to medical facilities [14,18,21]. For example, the elderly participants in the study by Heldal et al., who reported that they used to discuss with peers whether the issue leading to online searches needed to be followed up by a medical visit [16]. Another reason to avoid the doctor’s office was reported by one participant in the study by Silver et al., (2015) as he elaborated on distrust of his doctor and that he would rather look things up online [23]. Diviani et al., also registered how the participants applied online health information to solve health problems. They found that some patients actively modified their health behavior according to the online information, with or without discussing this with their medical contact. However, the behavior that was modified was mostly related to diet or physical activity, while it was never reported that they changed medication without discussing this with their GP (general practitioners) [14]. In the study by Longo et al. (2010) involving patients with long-term diabetes one of the patients reported how, if they were unhappy with the information from their health practitioner (HCP), they would keep searching information elsewhere until they found a satisfactory solution, and sometimes challenge the GP [19]. Protheroe and colleagues reported that patients were often reluctant to seek health information online due to information overload and lack of quality assurance. They further found that patients from privileged socioeconomic groups were more active in looking up information and taking health decisions, leading to better self-management and higher HL [22]. Although the internet was frequently used as information source in all the studies, there were patient -reports about information overload when searching for health information online, across all patient groups and ages.

#### 3.1.2. Calibrating One’s Own Understanding of Online Health Information with the Practitioner

Although most participants did use online health information, they shared some problems in terms of understanding and using the information. The authors mentioned examples of patients who thought it was better to receive information directly from a physician. Others went to the doctor to avoid unnecessary uncertainties [14]. Although many participants in Magasman-Conrad’s study described the internet as a useful tool for information, they emphasized that health care professionals were crucial to their health and treatment [20]. Health care professionals were also the most frequent and trusted source of information for the long-term diabetic patients in the study by Longo [19]. The connection to trust also emerged in Lee’s study, where one participant described how getting a list of various sources of information from their doctor was a valuable contribution. It gave them the freedom to choose between trusted websites, depending on what fitted their needs best [18]. Referrals to websites were also seen as positive by parents of sick children in a study by Neill et al. (2014) however, a need for support to understand the online information was also reported [21].

Other participants reported using online health information to prepare themselves before the medical appointment to be able to discuss issues with the doctor [14,15,18,24]. Some even educated themselves to be able to teach the doctor about their disease [14]. These behaviors were often combined with additional searches after the medical encounter. Silver et al., 2015, found three main facilitators for patients to be able to discuss online information with their doctor. In this case, having a family member present was beneficial. So were doctors who asked their patients about online health information or recommended various pages or digital tools to visit. A facilitator mentioned by one of the participants related to advertisements that suggested talking to your doctor. Participants who discussed their internet search with their doctor were more concerned about credibility or limitations of the online information, or limitations in their own ability to search and evaluate the information [23]. The participants in the study by Fahmer and colleagues, also reported that they checked the information they found online with professionals, as well as with peers, to check the quality of their information [15].

#### 3.1.3. Barriers to Sharing Online Health Information with the Practitioner

Some patients in the study by Diviani et al., reported negative experiences with discussing online information with their doctor, due to lack of acknowledgment, or rejection of the information that the patient had gathered. Others simply never discussed it with their doctor [14]. Silver et al., looked more closely at barriers that patients might have to communicating online information searches with their practitioners. The participants explained a sense of uncertainty in how to describe the information they had collected. They were also worried that discussing the online information would result in embarrassment, especially as they often found the online health information both endless and confusing [23]. These worries were described as implicit feelings the patients held about their own role and identity. However, others were worried about offending the doctor by talking about their online searches as “doctors do not want patients to tell them how to do their job” [23]. Some were also discouraged from discussing online health information with their doctors by younger family members, or the issue simply did not come up.

### 3.2. Practitioners’ Attitudes towards Online Information

Six interview studies concerning practitioners’ attitudes towards online health information were included in our study. Practitioners’ ambivalence towards information introduced by the patient from the Internet was reported by several researchers [13,17,25,26,27,28]. Caiata-Zufferey and Schulz [28] found that practitioners adopted four different strategies when meeting online informed patients: 1. Resistance to online information (neutralize information reported by the internet informed patient); 2. Repairing online information (correct the internet-informed patient and align the information to the doctor’s point of view); 3. Construction around online information (build a shared reality using the online information as a springboard); and 4. Enhancement of online information (empower the internet-informed patient by providing the instruments to obtain quality information. Which approach was used depended on how health literate they deemed their patients to be, in addition to their preconceptions about Internet information in general [28]. Woodward-Kron and colleagues discussed the role of online information in the physician-patient interaction through a triangulated model: The patient, the physician, and the internet. Their participants shared numerous factors that could influence how the physicians engaged their patients in online health information. The physician’s specialty mattered. However, the patients’ age, their family and how they engaged with internet-based information were also key factors. Other factors were the setting (they contextualized the work environment); for example, access to computers and information leaflets with URL addresses had to be available [27].

An older study by Hart et al., 2004, described a situation where practitioners worried more about their own online skills than their patients did, and also worried that the internet would encourage patients to challenge their health practitioners. However, it seems talking about online information with patients was less reflected upon in this early 2000s study [13]. In the Chinese study [17] Doctors reported that patients misinterpreting online information, could get in the way of treatment. Fredriksen, Moland [25] interviewed nurses, midwives, and doctors about their perceptions of internet/online health information in encounters with patients. They found that some practitioners felt that internet information was challenging their roles. With physicians having to adjust to competing information and nurses’ roles becoming more akin to coaching roles, researchers reported that practitioners were worried about becoming “unnecessary” [25]. The practitioners observed that online information was always indirectly present in the dialogue, although it was not always directly talked about [25]. The “unknown” knowledge, arriving from somewhere, was also described by Sjöström, Hörnsten [26]. Although many opportunities were mentioned, there were some ambivalences towards internet information, depending on how it was used (some patients predetermine their treatment and diagnosis, while others use it to understand and gain knowledge-this affects the consultation).

### 3.3. Other Findings

Albeit a limited sample of included studies, they demonstrated a variation in context-dependent aspects, such as socioeconomics, age, disease, culture and literacy. Two factors seemed to be relevant in all cases: the social network (family and friends), and whether health care professionals opened for talk about online health information.

Paternalistic views on the role-sets of the doctor-patient interaction could be held by both medical professionals and patients. Caiata-Zufferey and Schulz study suggested that practitioners took into account the health literacy level of the patients when they chose strategi in how to address online information [28]. Protheroe’s study reported that patients from lower socioeconomic groups rather took on a more passive role as patients, and as such incorporated the paternalistic view on the role of the GP [22]. There were also some participant reports stating slight gender imbalance regarding health information. For example, in the study by Magsamaen-Conrad it was reported that women (despite less digital skills) often where the ones to look up various health related issues online, to co-manage their partner or family-members health issues [20]. Similarly, the study by Nell et al., reported that in the parental situation, mothers were often the ones getting most access to information from the health care setting, and fathers could even be left without the hard-copy of the information [21].

## 4. Discussion

The objective of this paper was to get an overview of existing studies addressing how patients informed by online health information communicate about this with their practitioners, and to identify the key topics raised in these studies.

A systematic approach was taken to identify relevant qualitative studies in selected databases. A different approach may have resulted in different findings. The findings reflected the chosen inclusion and exclusion criteria, and included studies with a patient perspective and practitioners perspective, respectively.

The key topics raised in the studies with a patient perspective were (1) reasons to seek online information, (2) calibrating understanding of online information with health care practitioners, (3) barriers to discussing online information with health care practitioners. The studies suggested that patients had three major reasons for seeking online information: (1) self-diagnosing to avoid presenting at the doctor’s office and to take necessary measures to change their own health behavior, (2) to arrive prepared at the health care practitioner’s office, either to discuss or to challenge the health practitioner, (3) to supplement the information from the health practitioner, either because of unsatisfactory information or for quality assurance. While some used online information to appraise their own health condition and to actively try to avoid presenting to the doctor, others used the information to appraise their own condition, assumingly to be able to understand more about own condition in the discussion with the doctor. Patients calibrated their understanding of health information in the communication with the practitioners, and discussed to avoid uncertainties. This aspect of trust in the healthcare professionals was especially important for long-term patients, which is not surprising, as they are dependent on having good relationships with practitioners over time in their patient trajectory.

The studies explicitly or implicitly mentioned contextual resources that were important for the calibration of the patient’s understanding of health information, for example being advised about trustworthy online information versus advertisements, and by bringing family members to the consultation.

Barriers to sharing and discussing online health information with the health practitioner were primarily due to how the practitioner responded to the patient. Some patients reported uncertainty in how to convey the information they had read; they were unsure if they had understood the information correctly and did not want to embarrass themselves. Others avoided talking about online information in order to avoid embarrassing the practitioner by implicitly suggesting that he/she lacked knowledge.

The studies from the practitioner’s perspective suggested that there was ambivalence towards patients who wanted to discuss what they had read online about their condition. The study by Caiata-Zufferey and Schulz [28] interestingly pointed out that health practitioners had strategies for responding to, calibrating and contextualizing online health information. The contextualization was, for example, to enable the patient to understand their own health condition and to provide relevant online information of quality in order to empower the patient.

The topics from the studies mentioned above connect with the eHealth literacy aspect of how patients comprehend, appraise, and communicate online health information in various ways. First, patients are actively trying to *comprehend* online health information, and in some cases, this leads to self-diagnosing and avoidance of going to the practitioner. The patients who do not comprehend information by themselves prepare in order to be able to discuss it with the practitioner, or presumably they simply trust that the practitioner makes decisions on their behalf. Again, others bring family members to help with comprehension. In the process of comprehending, the patients would implicitly and explicitly *appraise* or assess whether the online information is understandable to the degree that they can act upon it in the given context with the available contextual resources. Finally, patients do communicate the online health information if the context allows it, for example if the practitioner is open to discussing the topic or responds to it in a way that does not embarrass the patient.

The results reflect patients who make use of online health information as active participants in their own health. They read up before and after doctor’s appointments, change health behavior and discuss this with peers. However, the input from practitioners in terms of quality web sites to search for information, discussing the information found online, and making well-informed decisions seems to be an important factor for the patient to both build trusted relationships with their practitioners, and to fully apply the available information to their own health benefit. The findings also give us brief insights in various cultural and social factors, like gender, socioeconomic situation, and societal development which can be of influence of how both patients and practitioners roles and responsibilities manifest in the interaction. Furthermore, there was only one study that used naturalistic, observational data [13]. In their material from 2004, the participants did not discuss online health information in their interaction. However, there is reason to believe that this have changed over the years, as we see in the interview studies that reflections regarding online health information, from both patient and practitioner point of view, have increased over the years. As such, it is important to assess and understand more about the approach that practitioners choose to take when addressing online informed patients in various contexts.

### Limitations of the Study

Our study has some limitations. Firstly, the sample size is quite limited (only 16 studies). A larger sample, with other inclusion criteria may have given other results. Secondly, the studies are published in English or in the Scandinavian languages. A broader approach may have yielded other perspectives. The conclusions might consequently be prone to sampling bias. Thirdly, patients with different health conditions might have different search behavior, so this study could have focused more on identifying studies addressing these differences.

## 5. Conclusions

Communication in clinical consultations is increasingly involving topics from internet information. eHealth literacy, that is, how patients comprehend, appraise and communicate online health information, is important for patient engagement and self-management, and consequently for promoting good health. Both patients and practitioners relate to online health information in the clinical encounters, but according to the interview studies reported here, online health information is given different value depending on the trust to the information, the person and the context.

Being a patient in the digitalized healthcare contexts is complicated. Various competencies and skills are needed to handle the contextual demands in the healthcare system and in society, as well as to cope with own illness. The patient must be able to access, critically evaluate and communicate health information, but he/she must also trust and relate to the communicative strategies of the practitioners. On the other hand, the practitioners’ practices are influenced by the institutional and professional frameworks and contexts as well as patient trends, such as patient empowerment and patient perspectives. When patients present to the consultations with questions and opinions about their own condition, the practitioners’ position, knowledge, and authority may be challenged. Practitioners, and health authorities, must take these contextual changes into account, and allocate both time and training to prepare for these communicative issues in the consultations with the patients.

More observational studies from authentic clinical communication situations are needed to capture how patients and practitioners deal with online health information in their dialogues, so we can understand more of what occurs in medical interactions in various social contexts.

## Figures and Tables

**Figure 1 ijerph-19-14004-f001:**
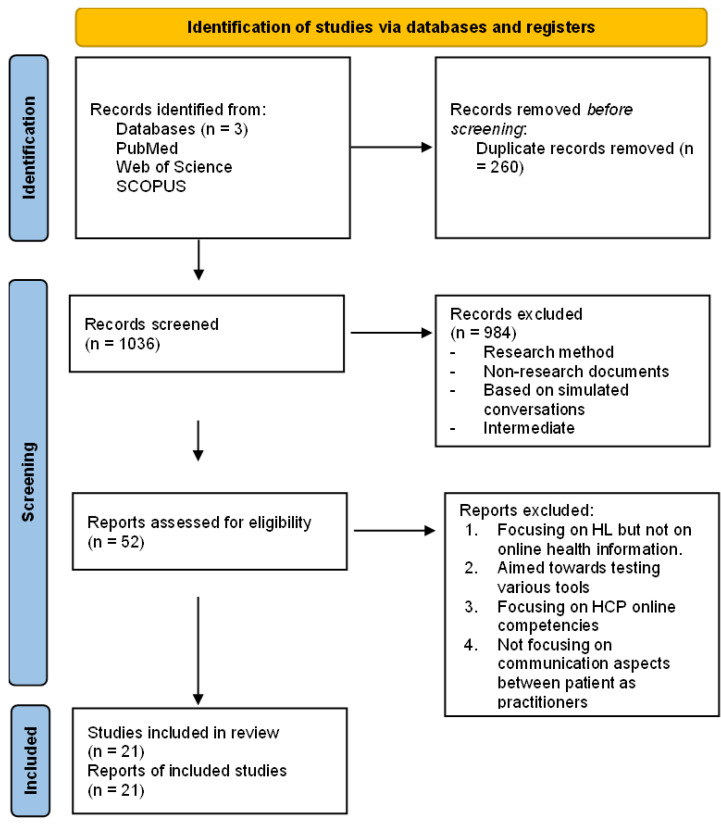
Prisma Flow-Chart [12].

## Data Availability

Not applicable.

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
