# Peer review of "Qualitative Research Studies Addressing Patient-Practitioner Communication about Online Health Information"

_ijerph, 2022, doi:10.3390/ijerph192114004_

Round 1
Reviewer 1 Report
This perspective paper presents a synthesis of published qualitative studies on patient-practitioner communications about online health information. Key topics have been identified from both patients and practitioners' perspectives. The paper is mostly well written and I have a few suggestions for the authors.
First, the sample size in the synthesis is quite limited (only 16 studies included). These studies are published only in English or Scandinavian languages. Therefore, any conclusions need to be carefully evaluated as they might be prone to sampling bias.
In addition, patients with different health conditions may have different online health information seeking behavior. As the author noted, there are patients with chronic conditions, gynecologic issues, and parents. While this study is largely a descriptive study, I feel that it is still worthwhile to explore how different types of patients differ in their health information seeking and communication with practitioners.
Lastly, a table summarizing the included 16 studies is necessary (can be supplementary information).
Author Response
Dear Reviewer 1.
Thank you for your feedback and suggestions for change. We appreciate your time.
Please see the attachment with a point-by-point response to yours and the other reviewers feedback. The reviewed document together with an overview table will be given for the re-submission.

Reviewer 2 Report
Dear authors,
The manuscript above describes nicely a systemic approach related to online health system. "The objective of this perspective paper was to get an overview of existing studies addressing how patients informed by online health information communicate about this with health care practitioners, and to identify the key topics raised in these studies".
A summarised table or graph with the main fundings of all studies is missing from the manuscript. A relevant table or graph should be added to improve the presentation and flow of manuscript.
Author Response
Dear reviewer 2
Thank you for your feedback and suggestions for change. We appreciate your time.
Please see the attachment with a point-by-point response to yours and the other reviewers feedback. The reviewed document together with an overview table will be given for the re-submission.

Reviewer 3 Report
The paper focuses on the interplay between eHealth literacy and the social context. The authors are interested in how patients comprehend, appraise and communicate online health information, before and during the encounter with the healthcare professional.
The paper is clear and well structured however the content is repetitive in some parts. As an example, the aim of the paper is repeated many times throughout the paper.
The whole text should be revised removing concepts-repetition where it is not needed.
Discussion and Conclusions sections shall be improved. Only stating “the need of additional observational studies to better understand how practitioners and patients deal with online health information in their dialogues and therefore what occurs in medical interactions in various social contexts” is not enough.
The authors should provide the reader with additional elements that may guide researchers to fill the gap that doesn’t enable a complete understanding of the investigated topic,
Moreover, the investigation process should have considered a classification phase of the analysed papers based on different criteria e.g., country of the study, public of private healthcare system, age of the people involved in the study, year of the study.
Some additional revisions and suggestions follow:
-Line 121: the research questions guiding this paper are: …
-Fig.1. The reported flow chart may include specific information regarding the inclusion and exclusion criteria and not only the numbers of the papers.
-Based on the reported information, 14 (12+2) papers are considered for patients’ perspective and 6 (4+2) for the GP’s one. Please revise the reported numbers on page 5 and 6.
Author Response
Dear reviewer 3.
Thank you for your feedback and suggestions for change. We appreciate your time.
Please see the attachment with a point-by-point response to yours and the other reviewers feedback. The reviewed document together with an overview table will be given for the re-submission.
